# Characterization of Complex Concentrated Alloys and Their Potential in Car Brake Manufacturing

**DOI:** 10.3390/ma16145067

**Published:** 2023-07-18

**Authors:** Ioana Anasiei, Dumitru Mitrica, Ioana-Cristina Badea, Beatrice-Adriana Șerban, Johannes Trapp, Andreas Storz, Ioan Carcea, Mihai Tudor Olaru, Marian Burada, Nicolae Constantin, Alexandru Cristian Matei, Ana-Maria Julieta Popescu, Mihai Ghiță

**Affiliations:** 1National R&D Institute for Non-Ferrous and Rare Metals, 102 Biruinței, 077145 Bucharest, Romania; ianasiei@imnr.ro (I.A.); dmitrica@imnr.ro (D.M.); cristina.banica@imnr.ro (I.-C.B.); mburada@imnr.ro (M.B.); alex.matei@imnr.ro (A.C.M.); mihai@imnr.ro (M.G.); 2Fraunhofer Institute for Manufacturing Technology and Advanced Materials—IFAM, Winterbergstr. 28, 01277 Dresden, Germany; johannes.trapp@ifam-dd.fraunhofer.de; 3Sigma Materials GmbH, Wupperstrasse 36a, 40699 Erkrath, Germany; andreas.storz@sigma-materials.de; 4Rancon S.R.L., 25 G. Coșbuc St., 70293 Iași, Romania; ioan.carcea@yahoo.com; 5Faculty of Materials Science and Engineering, University Politehnica of Bucharest, 313 Splaiul Independentei, 060042 Bucharest, Romania; nctin2014@yahoo.com; 6Romanian Academy, “Ilie Murgulescu” Institute of Physical Chemistry, 202 Splaiul Independenței, 060021 Bucharest, Romania; popescuamj@yahoo.com

**Keywords:** complex concentrated alloys, structural characterization, corrosion resistance, transportation applications, materials design

## Abstract

The paper studies new materials for brake disks used in car manufacturing. The materials used in the manufacturing of the brake disc must adapt and correlate with the challenges of current society. There is a tremendous interest in the development of a material that has high strength, good heat transfer, corrosion resistance and low density, in order to withstand high-breaking forces, high heat and various adverse environment. Low-density materials improve fuel efficiency and environmental impact. Complex concentrated alloys (CCA) are metallic element mixtures with multi-principal elements, which can respond promisingly to this challenge with their variety of properties. Several compositions were studied through thermodynamic criteria calculations (entropy of mixing, enthalpy of mixing, lambda coefficient, etc.) and CALPHAD modeling, in order to determine appropriate structures. The selected compositions were obtained in an induction furnace with a protective atmosphere and then subjected to an annealing process. Alloy samples presented uniform phase distribution, a high-melting temperature (over 1000 °C), high hardness (1000–1400 HV), good corrosion resistance in 3.5 wt.% NaCl solution (under 0.2 mm/year) and a low density (under 6 g/cm^3^).

## 1. Introduction

The braking system plays a crucial role in stopping or slowing down a vehicle by creating frictional resistance [1].

The braking system mainly consists of a disk bolted to the wheel hub and a caliper containing pads connected to the axle stationary housing. The caliper applies a braking force to the disk through the braking pads to slow down or stop the vehicle. The brake disk is the main component that makes the connection between the wheels and the car. High-mechanical forces are developing at this stage and friction produces high local heat (up to 1000 °C). Due to the importance and necessity of the brake disc, the material requirements have to be established and defined very clearly. High strength at low and higher temperatures, high-friction coefficient, good heat capacity and corrosion resistance are the main characteristics of brake disk materials [2]. On the other hand, the automotive industry is in continuous development, and it is important to consider a series of additional competitive factors, specifically, lightweight and low particulate emissions.

Several types of materials are used to manufacture the components of the braking system, such as cast iron, aluminum alloys, titanium alloys, composite materials and carbon fibers [3]. From all the materials, the most used is grey cast iron, due to its high strength, high-temperature stability and low-manufacturing costs [4]. Although grey cast iron has been used extensively, it has several disadvantages. Grey cast iron has a high density of 7.25 kg/cm^3^, and low-corrosion resistance, which determine high-fuel consumption, short maintenance intervals and significant particulate pollution. The most important disadvantage of using cast iron for the brake disk is environmental pollution. Recent studies show that 16–55 mass% of non-exhaust PM10 emissions are caused by brake wear [5]. Above all, size is an important aspect, as it determines how deep the particles can penetrate the human body. Using the PM10, PM2.5, PM1, and PM0.1 classifications, the average brake dust particle diameter is about 2 μm, well within the 2.5 range. Additionally, some sources show a second distribution peak around PM0.1, indicating the severity of the respirable dust and particulate matter problem. Small particles in the air can be inhaled and penetrate deep into the lung tissue, causing serious lung disease. Small particles are produced as a result of the disc and the brake pads friction. These particles remain in suspension in the atmosphere, being a source of pollution and a human health hazard [6,7,8,9]. Seo et al. [7] studied the suspended particle pollution of four types of brake discs made from three types of cast iron (FC170, FC200 FC250) and a ceramic material, in order to analyze the particle size and mass concentration. The study showed that cast iron generates the most airborne particles. Ghouri et al. [8] studied the influence of corrosion on particle pollution for grey cast iron disk brakes. It was found that corroded disks increase the particle emissions by double and also inhibited braking performance by reducing the coefficient of friction. The effect of temperature on particulate emissions was calculated by Seo et al. in [9]. It was found that the thermal conductivity is one of the main depending factors and that cast iron disks produce the highest amounts of particulate matter.

Complex concentrated alloys (CCAs) represent a promising solution to this challenge, due to the high versatility of their properties [10]. These new materials have the advantage of having a high number of component elements that can form complex combinations, which leads to a series of favorable properties. CCAs are including the high-entropy alloy (HEA) family by allowing a larger compositional domain, formation of intermetallic phases and without a minimum number of elements. Alloys from this group of materials generally have a higher configurational entropy than conventional alloys. Due to this characteristic, CCAs tend to form disordered and compositionally complex solid solution structures. The large number of component elements allows for a higher degree of freedom in alloy compositions and properties.

Several alloy compositions, containing low-density elements, have been investigated by several research teams [10,11,12,13,14]. In order to induce the high-entropy effect, some authors maintained the group of elements that are found in most of the studied HEA and added, in a controlled manner, lighter elements. A light weight HEA based on the Cr-Fe-Mn system, with gradual additions of Al and Ti, was studied by Feng et al. [11]. A dominant BCC structure was obtained at low additions of Ti; still, the alloy brittleness and relatively high density can be detrimental in several applications. Kushnerov and Bashev [12] investigated the Al and Si additions to the main Cu-Fe-Mn-Ni system and found that cooling rates strongly influence the phase constitution of the alloy. Most of the alloy compositions formed a single-phase FCC structure in the as-cast state, but the density of the material was over 4 g/cm^3^. A LWHEA with a high proportion of low-density elements was developed by TSENG et al. [13]. The alloy presented a predominant BCC-type solid solution structure and a lower level of intermetallic phases. The Al20Be20Fe10Si15Ti35 alloy delivered good mechanical properties and oxidation resistance. Still, Be is a safety hazard element and may raise the production costs. Recently, a research in the field of LWCCAs was performed by Gondhalekar [14], aiming to design and develop new light-weight aluminium based CCAs. Empirical rules from HEA design were applied in the process, alongside with thermodynamic modeling. Several compositions containing Ag were found to have good potential, but the alloy cost may restrict the use for light weight applications. Other compositions have been proposed by Mitrica et al. [15,16] but improvements need to be made for the reduction of intermetallic phase content and for particle distribution improvements.

The development of complex light alloys with higher strength and high-temperature stability has been researched by several authors.

A new alloy system, Al-Fe-Mn-Si, was designed and researched by O’Brien et al. [17], intended to deliver good corrosion resistance and mechanical properties. The new multiphase, CCA, has equimolar composition and is based on four principal elements, AlFeMnSi. The results showed that the developed alloy exhibited excellent corrosion resistance and high hardness due to the presence of Fe in high proportion; therefore, the alloy is less expensive with 46% than stainless steel 304 L, and has a lower density of 4.5 g/cm^3^.

The idea for the substitution of Ni with Ti for lowering the density in the well-known high-entropy alloy system, Al-Cr-Fe-Mn-Ni, was investigated by Rui Feng et al. in [11]. The authors investigated the phase formation in AlxCrFeMnTiy alloy system through the assessment of various empirical rules pertaining to light-weight HEAs. A comparison between experimental and modeling results showed significant differences. CALPHAD modeling was also performed on Al-Cr-Nb-Ti-V and Cr-Nb-Ti-V-Zr systems, producing enthalpies of mixing values at higher temperatures. 

A medium-entropy alloy (MEA) system, based on the Al-Ti-Cr-Mn-V system, was developed by Liao et al. [18] using a nonequiatiomic approach, maintaining Al concentration at 50 at.%, and the main group (TiCrMn) between 30 and 45 at.%. The results showed great mechanical properties before and after annealing, meaning a compression strength of 1940 MPa, while producing a ductility of 30%.

Several research pathways were followed in the past for replacing the well-known and widely applied cast-iron brakes. Different materials were trialed but they lack one of more of the cast iron main properties: high resistance, high-temperature stability and a high-friction coefficient. The aluminum matrix composite materials became the most promising materials, as they are relatively inexpensive to produce, have good mechanical resistance, a good friction coefficient, lower density and lower particulate emissions. The main impediment is the resistance at higher temperatures that is limited by the aluminum alloy matrix.

The present work focuses on the development of new alloys for the manufacturing of disk brakes with low density, high hardness, high-temperature stability and corrosion resistance. The main goal is to provide comparative analyses between two promising alloy systems, i.e., the modeling approach, and to investigate the influence of heat treatment on the structure and properties of the materials and the degree of predictability of the structure offered by the investigated methods. Another important aspect of this study is the applicability potential of the new materials to replace gray cast iron in the manufacture of the brake discs.

## 2. Materials and Methods

Obtaining a material that meets the requirements imposed by the transport industry and society requires a careful and elaborate selection between the possible CCAs systems. Considering that the particularities of each component element of the alloy have an impact on the final properties of the CCA, selected metals must have a positive effect on the desired characteristics. To obtain a material with a low density, elements such as aluminum, iron and silicon were selected. Cr, Fe and Mn have a relatively high density, but they can be used to increase the melting temperature of the alloy. Also, some of these elements, such as Al and Fe, have attracted attention due to their affordable cost and favorable contributions to the economic impact of the material. 

To obtain the most suitable composition of the alloys, there are several useful tools, including the semi-empirical criteria. The criteria with significant effects over the structure and the properties of the alloys are: mixing entropy (ΔS_mix_), enthalpy of mixing (ΔH_mix_), atomic size difference (δ), parameter Ω, Allen electronegativity difference (Δ_χ_), valence electron concentration (VEC) and the geometrical parameter (Λ). 

In this study, the degree of formation of solid solutions was analyzed by calculating Allen electronegativity difference and the geometrical parameter Λ. 

The parameter Λ has a high-predictive power on the alloy structure, as mentioned by Anil Kumar Singh et al. [19]. If it has a lower value than 0.24, it will form a multiphase structure containing intermetallic phases. Otherwise, if the aim is to obtain a structure with a single phase of solid solution type, then this parameter must have a value greater than 0.96. If Λ values fall between these two values, then two phase mixtures can be formed. 

The parameter Λ depends on the mixing entropy and the atomic size difference of the mixture. The mixing entropy is also an important parameter that characterizes the ability to form a solid solutions structure, and it is determined using Boltzmann’s equations [20]:(1)$$\Delta S_{mix}=-R\cdot \sum c_{i}\cdot lnc_{i}$$
where R is the gas constant and c_i_ is the molar fraction of the element i.

The atomic radius difference is considered the parameter with the strongest influence on the structure, and is preferable so that the component elements of the alloy have appropriate values [20]. The difference in atomic radius was calculated with Equation (2) [21].
(2)$$\delta =100\cdot \sqrt{\sum c_{i}\cdot {\left(1-\frac{r_{i}}{\bar{r}}\right)}^{2}}$$
where r_i_ is the atomic radius of the element i and $\bar{r}$ is the atomic radius average.

Knowing the difference in atomic radius and mixing entropy, the parameter Λ can be determined through Equation (3) [19]:(3)$$\Lambda =\Delta S_{mix}/\delta^{2}$$

To calculate the Allen electronegativity difference (Δχ), it is important to know the electronegativity after Pauling for element i (χ_i_) and the electronegativity average ($\bar{\chi }$). For the alloy to form solid solutions, its value must be between 3–6%. The Allen electronegativity difference was calculated using Equation (4) [22]:(4)$$\Delta \chi =100\cdot \sqrt{\sum c_{i}\cdot {\left(1-\frac{\chi_{i}}{\bar{\chi }}\right)}^{2}}$$

The thermodynamic and kinetic simulation of alloy structures is very useful in the optimization of CCAs systems. This was performed through the Matcalc Pro edition software, version 6.02. The CALPHAD method (CALculation of PHAse Diagrams) analysis is the basis of thermodynamic modeling, while the kinetic evaluation was achieved through the use of specialized modules that study solid-state phase transformations. 

The selected alloys were prepared using raw materials of technical purity and mixed to obtain a charge of 250 g of each individual alloy composition. The primary metals were Al, Fe, Mn and Si for the AlFeMnSi alloy, and Al, Cr, Fe, Mn and Ti for Al_4_CrFeMnTi_0.25_. The charge for each composition was placed in an alumina–zirconia crucible, in an induction furnace, Linn MFG, 300 type-, with argon atmosphere. After melting, the alloys were cast in a cylindrical copper mold and cooled in the furnace under vacuum. In order to increase the degree of homogeneity, the alloys were remelted several times. The as-cast samples were subjected to heat treatment in an electrical furnace, LHT 04/17 Nabertherm GMBH (Lilienthal, Germany). The heat treatment stage was conducted at 700 °C for 50 h with a slow cooling rate.

The obtained samples, before and after the heat treatment process, were characterized by chemical, structural, mechanical and corrosion analyses.

The chemical composition of the samples was analyzed by inductively coupled plasma spectrometry (ICP-OES) using an Agilent 725 spectrometer (Santa Clara, CA, USA). The structure of the samples was also investigated by SEM-EDAX characterization, with a scanning electron microscope, FEI Quanta 250 (FEI Europe B.V., Eindhoven, The Netherlands). It is equipped with an X-ray spectrometer (EDS), which provides information about the chemical composition of the sample phases. 

For a better understanding of the configuration of the phases, XRD analysis was also performed. The sample was analyzed by using a BRUKER D8 ADVANCE powder X-ray diffractometer (CuKα1 radiation, Johannson Ge (111) monochromator) equipped with a PSD Lynx-Eye detector in Θ–2Θ reflection configuration. Data acquisition was performed using DIFFRACplus XRD Commander (Bruker AXS) software in the 2Θ-Region 20–120 with step size 0.020, holding time 8.7 s/step, and sample rotation speed 15 rot/min. The powder sample was mounted on a zero background sample holder. The collected data were processed using a Bruker^®^ Diffracplus EVA Release 2018 software and the database ICDD^®^ Powder Diffraction File (PDF4+, 2019 edition) was used for the phase identification.

The corrosion tests were achieved using Voltalab 80 PGZ 402, with a Volta Master software, version 7.0.8. The tests were performed in a sodium chloride solution of 3.5%, according with the method presented in [23]. 

Designing an alloy that has the highest melting temperature is of great importance, because the higher it is, the higher the working temperature and the better the thermal resistance. The temperature at which the reactions with heat exchange took place were determined by the thermal analysis equipment Setsys Evolution, Setaram. 

Vickers microhardness of the samples were ascertained using a micro-indenter attachment Anton Paar MHT10 (Anton Paar, Graz, Austria, at 25 C, by an applied load of 2 N and a slope of 0.6 N/s. The microhardness was performed using a Zeiss Axio Imager A1m microscope (Zeiss, Jena, Germany).

## 3. Results and Discussions

### 3.1. CCAs Structure Design Depending on the Element Concentation

Considering the phase concentration variation in the AlCrFeMnTi and AlFeMnSi alloys systems, the Matcalc simulation program was used to study phase redistribution in the solidification process. 

The calculated phase distribution of the AlCrFeMnTi alloy system is shown in Figure 1. The variation of the elements has a significant influence on the phases that can form in the structure. At 200 °C temperature, the variation of eight phases was observed: BCC_A2, BCC_B2, Cub_A13, AlCr_2_, Al_2_Ti, Cr_3_Mn_5_, FeTi and H_Sigma. Figure 1a shows that a high concentration of BBC_B2 phase is found when the Al content exceeds 13 wt.%, while Cr_3_Mn_5_ and Cub_A13 phases suffer a decrease in phase fraction. Increasing the aluminum content in the alloy does not have a strong influence on obtaining the AlCr_2_ phase, which reaches a peak of 0.05 phase fraction between 11 wt.% and 13%. It was also observed that the increase in the Al content leads to a decrease in the concentration of the BCC_A2 and FeTi phases, and after 15 wt.% Al these phases are not found any more in the structure. On the other hand, from Figure 1b, it can be observed that the BCC_B2 phase decreases in proportion to the increase in Cr content. The solid solution BCC_A2 reaches the peak of 0.2 phase fraction at 12%Cr. To obtain a structure composed of the intermetallic compound Cr_3_Mn_5_, it is necessary for the Cr content to exceed 11%. Between 12 wt.% and 17 wt.%, the variation of the AlCr_2_ phase can be observed. The proportion of the H_Sigma phase increases after 15 wt.% Al, while FeTi increases after 22 wt.%. The variation of Cr in the alloy does not have a significant influence on FeTi and Cub_A13, which no longer appear in the structure after 1 wt.%, and, respectively, 7 wt.%. The variation of Fe on the phases can be seen in Figure 1c. As in the case of Cr, increasing the Fe content leads to a reduction in the proportion of the BCC_B2 phase, which reaches a peak of 0.8 phase fraction at 10 wt.% Fe. A significant proportion of the BCC_A2 phase can be observed between 13 wt.% and 19 wt.% Fe. This element has an inconstant influence on the AlCr_2_ and Cr_3_Mn_5_ phases, noting that at 15 wt.% Fe, the two phases no longer appear. On the other hand, FeTi and H_Sigma phases increased in proportion after 25 wt.% Fe and, respectively, 18 wt.% Fe. In Figure 1d, it can be observed that the presence of Mn influenced the formation of the phase’s structure by favoring the increasing of the BCC_B2 phase, until 24 wt.% Mn. Also, after 23 wt.% Mn, the FeTi phase starts to increase in proportion. When up to 13% wt.% Cr, the Al_2_Ti phase decreases in concentration, while the H_Sigma phase increases. The AlCr_2_ phase reaches a peak of 0.3 phase fraction at 13 wt.% Mn, after which it decreases. Figure 1e shows the influence of Ti concentration on the phases variation. BCC_B2 is the predominant phase in the structure at a content of less than 20 wt.% Mn. The BCC_A2 solid solution is found in high proportions in the alloy structure at a low Ti content, but after 13 wt.%, this will no longer be present. On the other hand, after 13 wt.% Ti, the H_Sigma phase increases in proportion, reaching approximately 0.25 phase fraction at 16 wt.% Ti. The AlCr_2_ phase is observed to reach 0.2 phase fraction at 15 wt.% Ti, after which, it decreases in proportion. On the other hand, it was observed that the Cr_3_Mn_5_ phase has an inconstant growth with the increase of Ti.

The phase evolution, depending on element concentration, was calculated for AlFeMnSi alloy system (Figure 2). A higher concentration of Al has a strong influence over the formation of Al_2_Fe, which becomes more stable over 15 wt.% Al (Figure 2a). The influence of Al content is shown also by the transition between Al_2_Fe and BCC_B2 at higher Al concentrations. After 26% wt.% Al, Al_2_Fe phase increases in proportion, while the other phases decrease. The aluminum concentration has a minor influence on the formation of the BCC_A2_#01 phase: a variation of the BCC_A2 phase.

The proportion of the FeSi phase suffers a major decrease at 26 wt.% Al. Figure 2b shows the influence of Fe content over the formation of solid solution and intermetallic compound phases in the selected alloy. The proportion of the solid solution phases varies with the increase of Fe content; a transition between BCC_B2 and BCC_B2#01 phases occurred at 26 wt.% Al. Another transition was observed at 16%wt.% Al, between BCC_A2_#01 and Al_2_Fe. The presence of Fe has a significant contribution to the stability of FeSi phase, which reaches a peak of 0.5 phase fraction at 35 wt.% Fe. By increasing the Fe content, the concentration of Al_2_Fe has the higher peak of 0.6 at 25 wt.% and then decreases. The alloy structure is also defined by the presence of Mn, which has an important contribution to the stabilization of the complex compound-based phases (Figure 2c). Increasing Mn proportion has an important influence on the stability of BCC_B2 and FeSi. Si content has a strong influence over the formation of the FeSi phase, which reaches the highest peak at 30 wt.% Si (Figure 2d). This element content has also a high influence over the formation of the BCC_B2 solid solution phase, which is inversely proportional to the increase of the component proportion. On the other hand, the Al_2_Fe phase is insignificantly affected by the silicon content.

### 3.2. Thermodynamic and Kinetic Criteria Calculation

In the case of CCAs, the elements’ proportion has a high impact on the alloy structure. Based on the criteria calculation results, the optimal composition of the selected alloy’s systems can be designed, varying the proportions of each element from 0.5 to 2.5 molar concentration. Table 1 shows the criteria calculation for the AlCrFeMnTi alloy system and Table 2 for AlFeMnSi alloy system.

Figure 3 and Figure 4 illustrate the influence of each element composition on the most representative parameters for the AlCrFeMnTi and AlFeMnSi alloy system. The optimal range for solid solution formation is represented by Λ > 24 J/mol·K and Δχ < 6%, as it is presented in Section 2. To design a material with potential application in the manufacture of the brake disc, the influence of the elements compositions on density and melting temperature was analyzed. The dotted line in the diagrams represent the range limit.

The relationship between the parameter Λ and density for the AlCrFEMnTi alloy system is shown in Figure 3a. Al does not have a significant influence on the formation of solid solution phases. On the other hand, the use of aluminum in high concentration contributes to reducing the density. By analyzing the graphic representation of Al influence, it was observed that the melting temperature is inversely proportional to the increase in aluminum content (Figure 3c,d). It was also observed that Al has a positive influence on the alloy electronegativity (Figure 3b,d). Ti has a positive influence on decreasing the alloy density and improves the resistance to high temperatures. Analyzing the variation of Λ with density and with melting temperature, it can be observed that Ti is not a good solid solution former. Cr and Fe influence the structural characteristics of the alloy in a similar way, contributing favorably to the increase of the melting temperature. These elements are rather heavy and increase the alloy density, considerably. It was observed that increasing the proportions of Cr and Fe has a negative influence on the formation of solid solutions. On the other hand, it was observed that increasing the Mn content influences, positively, the formation of the solid solution phases. But it also affects the increasing of the alloy density and reduces the melting temperature.

Figure 4a shows the ratio between the alloy density and the parameter Λ, where Mn and Fe have a positive influence on the formation of solid solution. The tendency of forming solid solutions was observed by analyzing the ratio between electronegativity and density and melting temperature (Figure 4b,d). Mn and Fe are elements with a moderate melting temperature and increasing their concentration contributes to obtaining an alloy with potential application on manufacturing brake discs (Figure 4c,d). Increasing Al and Si content does not stimulate the formation of a solid solution, as observed by analyzing the parameter Λ and Δχ. An advantage of these elements is that they are light, which has a positive effect on reducing the density of the alloy

### 3.3. CCA Selection Using CALPHAD Method and Criteria Calculation

The information obtained through the Calphad method, regarding the complex materials design and optimization, establish an efficient approach to determine multicomponent phase diagrams using the mathematical determination of Gibbs free energies and diffusion mobility for each system phase [24]. The phase stability parameters and the effect of the composition on the phase formation are determined by the Calphad calculated composition-temperature diagram for the alloy. Therefore, the relation between the empirical criteria and the alloy composition is highlighted by the interpretation of those type of diagrams [25].

By analyzing the CALPHAD modeling results for the AlCrFeMnTi alloy system it was observed that a high percentage of Al contributes to obtaining a structure based on solid solutions. To obtain a suitable material for the brake disc, the density, melting temperature and the criticality of the elements were considered in the selection process.

The criteria calculation results provide information about the positive and negative influence of the elements over the formation of solid solutions. For the AlCrFeMnTi alloy system, it was observed that Mn has a favorable effect on the evolution of the parameter Λ. Regarding the Allen electronegativity difference, the increase of Cr, Fe and Mn tend to bring the alloy to the optimal zone. For the material to have potential application in the manufacture of the brake disc, the increasing of Al content contributes to the reduction of alloy density. Ti has the same effect over the alloy weight and improves the melting temperature. In the selection process, the reduction of its content was taken into account, because Ti is a critical raw material.

The CALPHAD modeling and the empirical calculation show that the AlCrFeMnTi alloy system has a promising capability to form structures preponderantly based on solid solutions. It was observed that a high content of Al favors the formation of BCC_B2, while Cr, Fe and Ti have a strong influence over the formation of BCC_A2. On the other hand, the criteria calculation results show that a high percentage of Fe and Mn has a favorable influence over the formation of a solid solution structure. Considering that the material weight is an important characteristic of the brake disk, Al content is increased to reduce the alloy density. Ti has positive influence over the alloy structure, but is using a lower percentage because it is a critical material. Several preliminary trials were made in order to meet all the proposed requirements. After analyzing the modeling and the preliminary trials results, a complex concentrated alloy was selected: Al_4_CrFeMnTi_0.25_ (Table 3).

The CALPHAD modeling results of AlFeMnSi alloy system show that a high percentage of Fe and Mn contributes to the obtaining of a preponderantly based on a solid solution structure. Otherwise, the Al and Si have a negative influence over the formation of hard intermetallic compounds: FeSi and Al_2_Fe.

Criteria calculation provide useful information for the selection of the alloy compositions. For the AlFeMnSi alloy system, Fe and Mn have a positive influence on the evolution of the parameter Λ and the Allen electronegativity difference. The increasing of these elements also contributes to obtaining a higher melting temperature. On the other hand, Al and Si have a favorable influence over the alloy density.

The modeling results of the AlFeMnSi alloy system show that the system can provide alloys with majorly solid solution structures. The CALPHAD modeling and the criteria calculation show that high percentages of Mn and Fe favors the formation of the solid solution structure, BCC_B2, while increasing Al and Si content has a negative influence on the formation of intermetallic compounds. Because the material weight is an important requirement of the brake disc, Al’s and Si’s influence over the alloy density allows to analyze the selected alloy’s compositions. A suitable composition was selected in order to satisfy the proposed requirements of the brake disc after several preliminary trials. The selected composition is AlFeMnSi (Table 3).

In order to predict the behavior of the selected composition in specific solidification conditions, the equilibrium and Scheil–Gulliver diagrams of the alloys (Figure 5 and Figure 6) were calculated. In the Al_4_CrFeMnTi_0.25_ phase diagram (Figure 5), it can be observed that the complex concentrated alloy has a high content of BCC_B2, Al_8_Fe_3_, AlCr_2_ and Al_8_Mn_5__D810 phases at room temperature. Also, the intermetallic compound Al_3_Ti_L is also present in the structure, but in a smaller proportion. The equilibrium diagram shows that the BCC_B2 phase has a higher concentration in the structure until approximately 750 °C, after which, it decreases. After the temperature of 100 °C, the BCC_A2 solid solution becomes the majority, reaching approximately 0.6 phase fraction, but it reduces significantly at almost 750 °C. The Al_5_Fe_2_-based phase has a high proportion between 300 and 500 °C.

The non-equilibrium solidification phases predicted by the Scheil–Gulliver diagram are presented in Figure 6. The phases with S termination on the name define the equilibrium and non-equilibrium values of the cumulative solidification patterns. It can be observed that the BCC_B2 phase is formed first during the solidification process. After, it can be observed that the BCC_A2, Al_5_Fe_2_ and Al_13_Fe_4_ phases are formed, which are stable. The Al_3_Ti_L, AlCr_2_, Al_8_Fe_3_ and Al_8_Mn_5__D810 phases do not show up in the Scheil-Gulliver diagram, since their formation occurs at temperatures below 650 °C. In the Scheil-Gulliver diagram of the Al_4_CrFeMnTi_0.25_ alloy, the melting temperature is lower than in the equilibrium phase diagram (Figure 5).

The equilibrium diagram of AlFeMnSi alloy shows the formation of three phases: BCC_B2, Al_2_Fe and FeSi (Figure 7). BCC_B2 and FeSi are the preponderant phases until the melting process begins at 800 °C. The equilibrium diagram shows that the Al_2_Fe phase has a lower concentration and decreases significantly after approximately 500 °C. The Scheil-Gulliver diagram provides useful information regarding the behavior of the alloy upon solidification (Figure 8). The non-equilibrium solidification shows differences between solidification temperatures of the phases. The FeSi phase solidifies first in the AlFeMnSi alloy, followed by the BCC_B2 phase, which has the solidification starting point at around 930 °C. The Al_2_Fe phase is not shown in the Scheil diagram, as its formation occurs at lower temperatures.

### 3.4. The Experimental Results of the Studied Alloys

The as-cast and annealed samples were chemically, thermally, and structurally analyzed in order to reveal the experimental characteristics. Corrosion tests were also performed on the resulting alloys’ specimens, along with density and microhardness measurements.

The chemical analysis of the samples in the as-cast state demonstrated that the products were obtained with a composition close to the nominal one (Table 4).

The optical micrograph analysis showed that there are no significant differences between the as-cast and quenched states (Figure 9). Usually, after solidification, the alloys present a dendritic-type structure, while in the case of the Al_4_CrFeMnTi_0.25_ as-cast alloy, it is observed that the structure has a polygonal aspect (Figure 9a). The grains are well defined, equiaxial in shape, and finer in the marginal area than in the central part. The thermal treatment implied the rapid cooling of the structure, which inevitably led to the reduction of the grain size, which can be seen in Figure 9b.

Figure 10 shows the scanning electron microscopy results where it can be observed that there are no differences regarding the morphology of the structure or the number of phases. The as-cast alloy (Figure 10a) has a polygonal structure with finer grains in the marginal area. After analyzing the SEM results of the cast alloy, hard lamellar phases were observed inside the grains. After the thermal treatment, the polygonal structure is preserved, as can be seen from Figure 10b. On the other hand, the lamellar hard phase is better distributed inside the grains. The EDS-mapping analysis (Figure 11 and Figure 12), performed on the two states of the alloy, provided new information about the structure of the material, especially about the distribution of the elements in the phase. For the as-cast alloy, it was observed that the elements are distributed approximately uniformly.

Regarding the composition of the lamellar phase, it can be observed that it has a composition similar to that of the matrix phase. However, in the case of the heat-treated alloy, it was observed that the distribution of the elements is mostly uniform, with the exception of Ti, which is concentrated in the marginal area of the grains (Figure 12).

The phases of the as-cast alloy were studied using the XRD method and it is shown in Figure 13. The results of the analysis showed that the structure of the as-cast alloy is mostly composed of a D8_10_ type phase, which is an intermetallic compound (Al_14.79_Mn_10.71_) that crystallizes in the trigonal system. The as-cast sample also contains a solid solution phase, BCC_B2 type (Ti_1.04_Fe_0.862_Mn_0.096_) and a D0_22_ type phase (Al_3_Ti) in reduced proportions. The heat-treated sample consists of the same phases as the cast one: D8_10,_ BCC_B2 and D0_22_.

The determination of the phase transformations and the melting temperature for Al_4_CrFeMnTi_0.25_ alloy was determined by differential thermal analysis. Figure 14 shows an exothermic phase transformation that takes place between 1000 and 1050 °C. But the solid–liquid transformation takes place at a higher temperature, between 1100–1200 °C. During the modeling stage, it was calculated that the melting temperature of the alloy would be approximately 1400 °C, but it was experimentally proven that the material melts at a lower temperature.

Optical analyses of the as-cast and quenched AlFeMnSi alloy samples (Figure 15) revealed that there are no significant differences between the structure of the two states. The as-cast sample has two phases, one dendritic and one interdendritic, well delimited between them (Figure 15a). The structure of the alloy does not show significant changes following the quenched treatment, which can be seen in Figure 15b. But the results for the optical micrographs analyses of as-cast and quenched samples showed that the application of thermal treatment contributed to the reduction of the grain size.

The dendritic structure of the as-cast AlFeMnSi alloy was highlighted by SEM analysis (Figure 16a). The interdendritic and dendritic phases are well defined, which can also be observed in the case of the quenched sample (Figure 16b). The EDS-mapping results show no large differences in terms of element distribution (Figure 17 and Figure 18). The as-cast alloy presented a dendritic structure with a higher concentration of Fe, Mn and Si. Al was observed to compose the majority in the interdendritic area. The two phases are well defined, which was also observed after the analysis of the quenched sample. EDS-mapping analysis showed that there are significant differences regarding the size of the phases.

XRD phase analysis of the as-cast and heat treated (Figure 19) AlFeMnSi alloys demonstrated that the samples have similar structure configuration, showing two phases: BCC_B2 and B20 types. Analyzing the results, it was observed that the as-cast sample is mostly made up of the B20 phase, while the BCC_B2 solid solution is in a smaller proportion. After quenching and natural aging, there were no differences in terms of the type of phases or their number, only the B20 phase increased in proportion.

Differential thermal analysis offers information about the temperature at which a phase change occurs. From the theoretical calculation, the melting temperature of the AlFeMnSi alloy was predicted to be around 1200 °C, but the thermal analysis shows that the solid–liquid transformation is between 1026 and 1074 °C (Figure 20).

Heat treatment has been one of the most convenient and economical ways of enhancing the mechanical properties of a material. In the case of aluminum alloys, it was observed that the application of a natural aging treatment after quenching contributes significantly to the increase of the hardness. For the AlFeMnSi alloy, it was observed that the sample increases in hardness after the application of quenching and natural aging by approximately 15% (Table 5). But for the Al_4_CrFeMnTi_0.25_ alloy, a decrease in hardness by more than 10% was observed. Although after the thermal treatment no significant differences were observed, the type of phases and their number remained the same; the redistribution of the elements led to the modification of the hardness of the material.

Corrosion resistance is an important property for the material used in the manufacturing of brake discs. The samples were tested by potentiodynamic polarization measurements (linear polarization resistance (LPR), Tafel plots) and performed in an aerated 3.5 wt.% NaCl solution. For a better interpretation, the results were compared with those of grey cast iron, the material that is usually used in the manufacturing of brake discs. Table 6 shows the corrosion potential (E_corr_), corrosion current density (i_corr_), polarization resistance (R_p_) and corrosion rate (CR). It is observed that the values obtained for the CCA alloys are superior to the conventional cast iron material. The heat treatment process has little influence on the corrosion resistance for Al_4_CrFeMnTi_0.25_ alloy but has a larger influence on the AlFeMnSi alloy. The values obtained of 0.013 mmpy and 0.003 mmpy, respectively, represent great improvements in materials used for brake disc manufacturing.

## 4. Conclusions

This paper presents the selection and characterization process of two CCA alloys to determine their potential use in brake disc material manufacturing. In the current study, the AlFeMnSi and AlCrFeMnTi alloy systems were proposed. To design the optimal composition, the Matcalc program was used, which provided information about the redistribution of solid solutions during the solidification process. Also, to determine the right composition, the thermodynamic and kinetic criteria were calculated, varying the concentration of each element.

In the case of the AlCrFeMnTi alloy system, it was found that the density decrease and the formation of the BCC_B2 solid solution are favored by increasing the Al content. The increase in the concentration of Cr in the alloy contributes to the increase in the phase fraction of intermetallic compounds, such as FeTi and Cr3Mn5, but also to the decrease in the proportion of solid solutions. A concentration of Cr and Fe higher than 20 wt.% leads to a reduction in the proportion of the BCC_B2 solid solution. Also, increasing the Cr and Fe content increases the density and melting temperature of the alloy. Mn has a significant influence on the increase in density, melting temperature and the formation of solid solutions. Increasing the Ti content has a positive effect on the density and melting temperature of the alloy, but does not facilitate the formation of solid solutions. However, at a content lower than 20 wt.% Ti, the phase fraction for the BCC_B2 phase is high.

Several possible compositions were tried to determine an alloy with high potential to be used in the manufacture of the brake disc. The selected complex concentrated alloy was: Al_4_CrFeMnTi_0.25_.

In order to predict the type of phases and their number for the Al_4_CrFeMnTi_0.25_ alloy, the equilibrium diagram was created. The modeling results predicted that the alloy structure at ambient temperature will consist of five phases: Al_8_Fe_3_, BCC_B2, AlCr_2_, Al_8_Mn_5__D810 and Al_3_Ti_L. The XRD analysis showed that the structure of the cast sample consists of only three phases, but these were also found in the equilibrium diagram: D810, BCC_B2 and Al_3_Ti. Modeling by the CALPHAD method provides indicative information about the possible phases that can form, which means that there may be differences between the predicted results and those obtained experimentally. Also, the database that was used is not specialized on multi-component alloys.

The two selected alloys were obtained using an induction furnace. In order to improve the mechanical and physical characteristics, the samples were subjected to an annealing process.

Following the optical, SEM and XRD analysis of the Al_4_CrFeMnTi_0.25_ alloy (as-cast and annealed) and AlFeMnSi alloy (as-cast and annealed) samples, the presence of solid solution phases and intermetallic compounds was observed.

The Al_4_CrFeMnTi_0.25 a_s-cast alloy is characterized by a polygonal structure, which is preserved even after applying the natural aging treatment, but a decrease in the size of the grains is observed. Following the ESD analysis, it was found that the constituent phases are uniformly dispersed in the structure with an agglomeration of titanium in the marginal area of the grains. After the simulation through the Matcalc program, it resulted that three phases will form in the alloy. Through XRD analysis, the presence of a BCC-type solid solution phase was observed, which was also determined through modeling. Through the Scheill–Gulliver diagram for the Al_4_CrFeMnTi_0.25_ alloy, it was found that the BCC_A2 phase is formed after the solidification of the alloy and that it is a stable one. Following the simulation obtained through the Matcalc program, a composition close to the empirical one was not obtained in terms of the phases of the intermetallic compound. Following the structural analysis, the results were: Mn_6.32_Al_6.68_ and Mn_10_Al_16_. The melting temperature of the alloy is approximately 1200 °C, but following the thermal analysis a phase change of around 1000 °C was observed. The thermal aging treatment influenced the mechanical characteristics of the sample, resulting in a decrease in microhardness after the analysis. And corrosion resistance was influenced by natural aging, but not significantly.

Through optical microscopy and SEM, it was observed that the AlFeMnSi alloy has a dendritic structure. In the interdendritic area, the presence of acicular compounds was highlighted. This structure showed slight changes after the application of the thermal treatment, but a reduction of the grain sizes was observed. The EDS analysis showed an increase in the number of phases in the alloy after the natural aging thermal treatment. The modeling showed that the alloy will have four constituent phases: BCC_B2, BCC_B2_#01, Al_2_Fe and FeSi. The empirical results showed that the cast alloy has two main phases: BCC (Fe_0.95_Al_0.89_Si_0.16_) and cubic (MnAl_0.05_Si_0.95_). The heat treated phase distribution showed a smaller proportion of the BCC phase. The samples were also thermally analyzed. After the analysis, it was determined that the AlFeMnSi alloy has a melting temperature of approximately 1000 °C, more than what was predicted by the equilibrium diagram obtained through the Matcalc program. In comparison with the previously studied alloy, an increase in the microhardness of the sample was observed for the AlFeMnSi alloy after applying the heat treatment. Also, corrosion resistance has improved and is superior to cast iron.

The two studied alloys presented great properties and have the potential to be used in the manufacture of brake disc material. However, further studies are necessary for the development of the disk brakes based on these materials, where reliable alloy production, processing and process integration are crucial.

## Figures and Tables

**Figure 1 materials-16-05067-f001:**
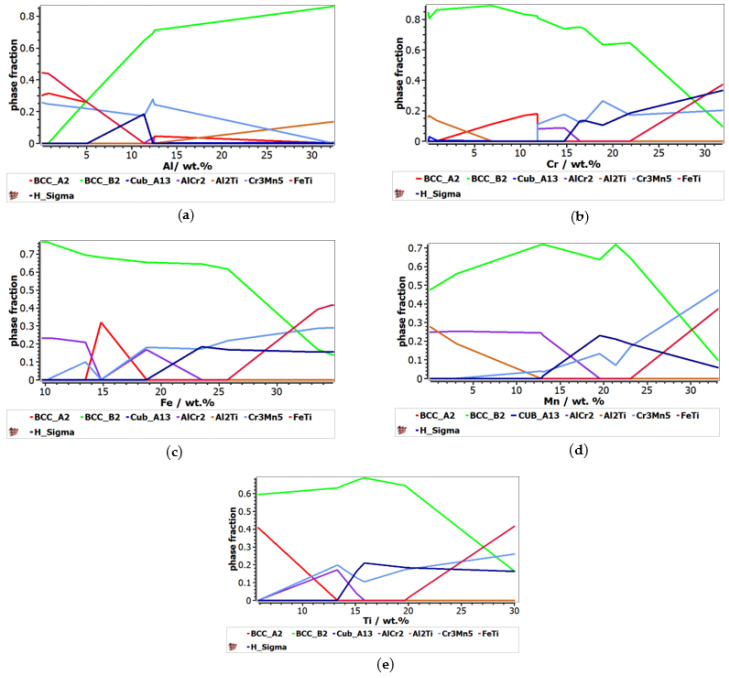
The impact of the alloying element on the phase stability for the AlCrFeMnTi alloy system: (**a**) Al influence; (**b**) Cr influence; (**c**) Fe influence; (**d**) Mn influence; (**e**) Ti influence.

**Figure 2 materials-16-05067-f002:**
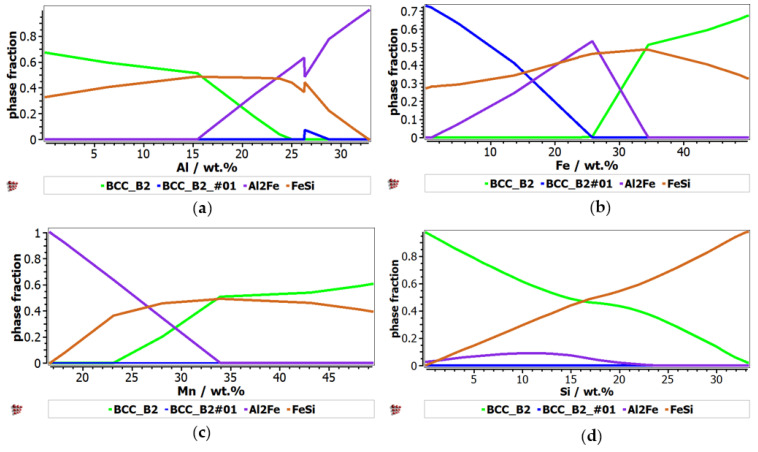
The proportions of phases that can be found in the Al-Fe-Mn-Si alloy system, at 400 °C, depending on the variation of the component elements: (**a**) Al; (**b**) Fe (**c**) Mn; (**d**) Si.

**Figure 3 materials-16-05067-f003:**
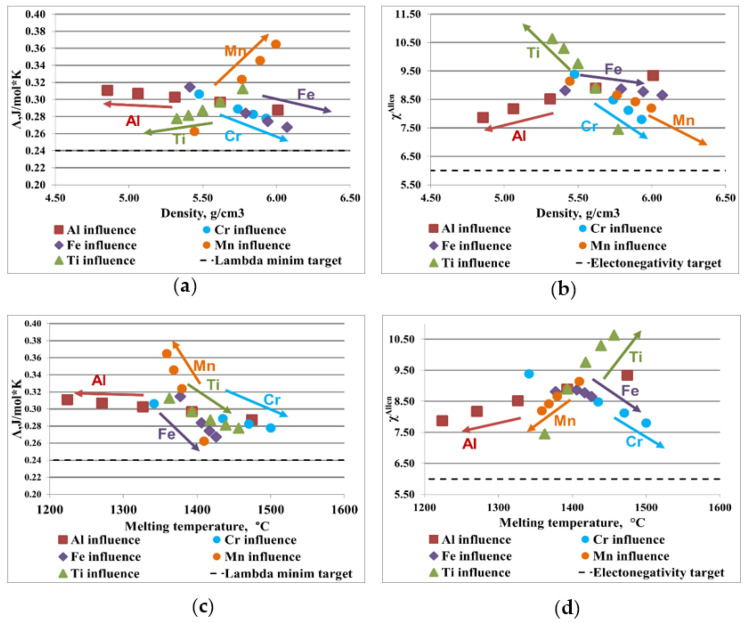
Graphical representation of the elemental influence over the criteria values for AlCrFeMnTi alloy system: (**a**) density, ρ, and the parameter Λ; (**b**) density, ρ, and the Allen electronegativity difference, Δχ; (**c**) the melting temperature and the parameter Λ; (**d**) the melting temperature and Allen electronegativity difference, Δχ.

**Figure 4 materials-16-05067-f004:**
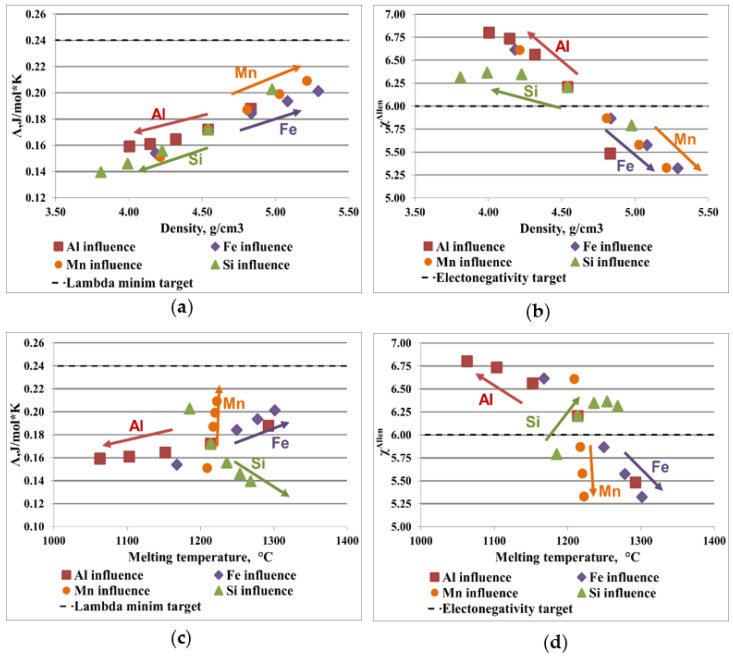
Graphical representation of the elemental influence over the criteria values for AlFeMnSi alloy system: (**a**) density, ρ, and the parameter Λ; (**b**) density, ρ, and the Allen electronegativity difference, Δχ; (**c**) the melting temperature and the parameter Λ; (**d**) the melting temperature and Allen electronegativity difference, Δχ.

**Figure 5 materials-16-05067-f005:**
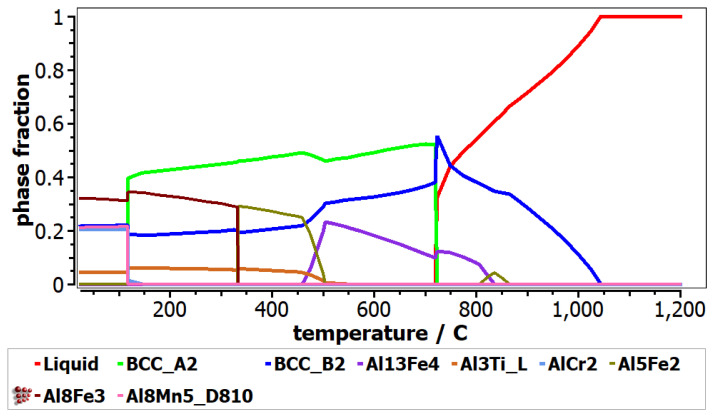
Equilibrium diagrams for Al_4_CrFeMnTi_0.25_ alloy.

**Figure 6 materials-16-05067-f006:**
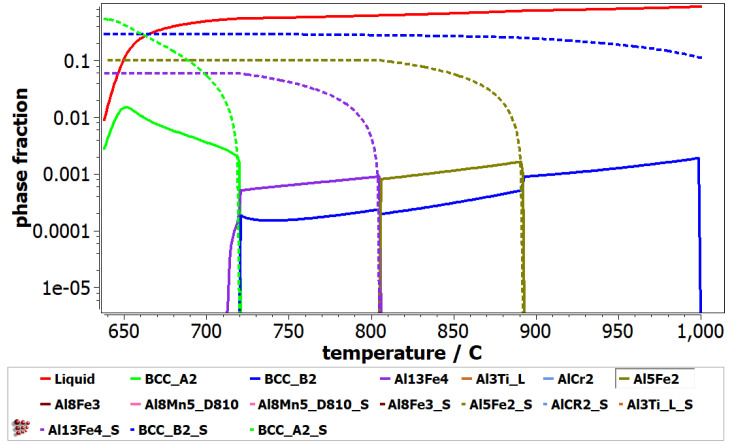
Scheill-Gulliver diagram for Al_4_CrFeMnTi_0.25_ alloy.

**Figure 7 materials-16-05067-f007:**
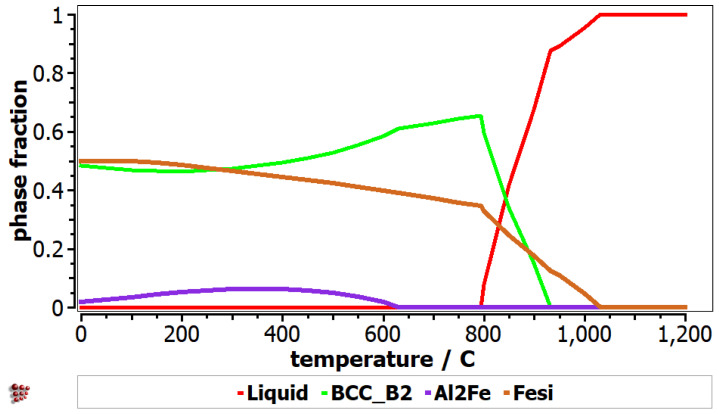
Equilibrium diagram for AlFeMnSi alloy.

**Figure 8 materials-16-05067-f008:**
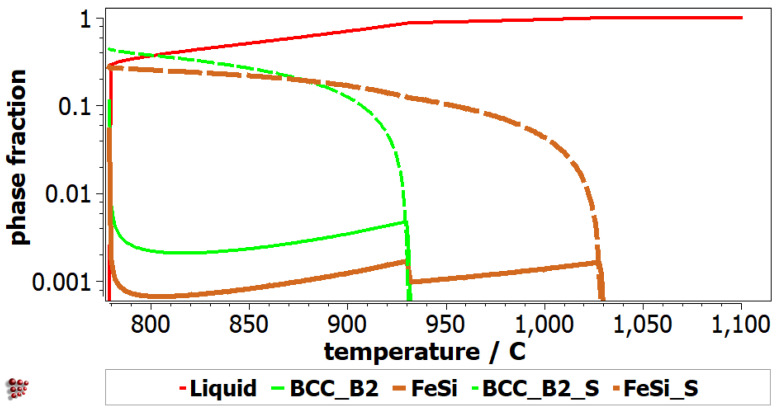
Scheill-Gulliver diagram for AlFeMnSi alloy.

**Figure 9 materials-16-05067-f009:**
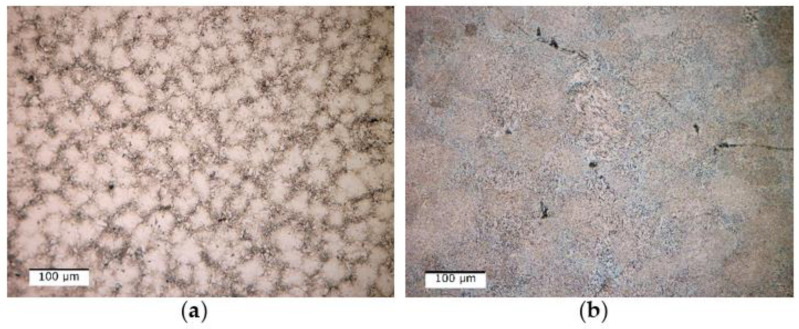
Optical micrographs of (**a**) as-cast and (**b**) quenched Al_4_CrFeMnTi_0.25_ alloy.

**Figure 10 materials-16-05067-f010:**
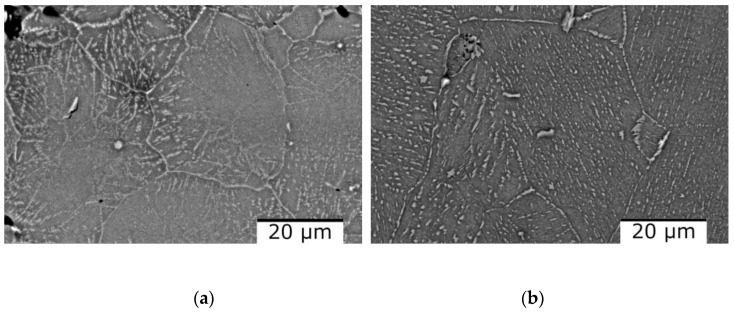
SEM images of the Al_4_CrFeMnTi_0.25_ alloy in as-cast (**a**) and heat treated (**b**) states.

**Figure 11 materials-16-05067-f011:**
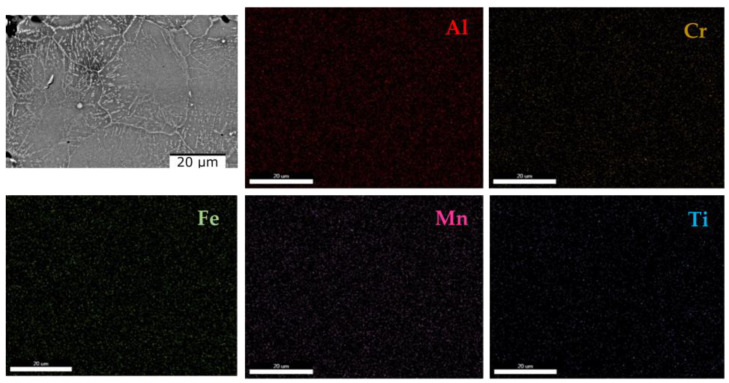
EDS mapping of the as-cast Al_4_CrFeMnTi_0.25_ alloy.

**Figure 12 materials-16-05067-f012:**
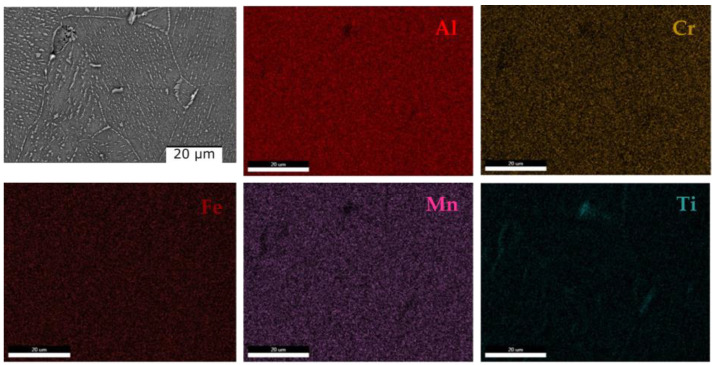
EDS mapping of the quenched Al_4_CrFeMnTi_0.25_ alloy.

**Figure 13 materials-16-05067-f013:**
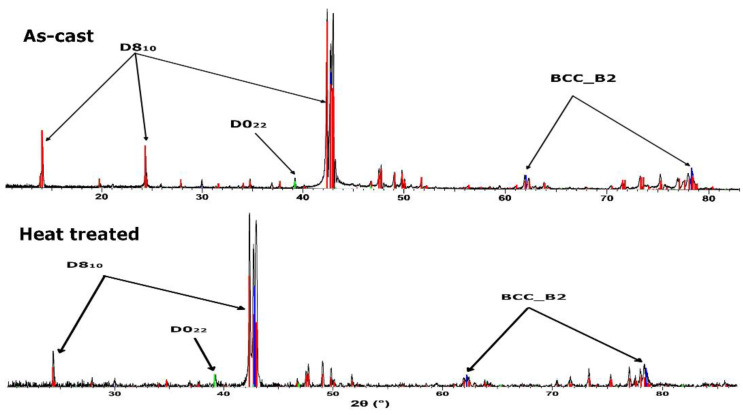
X-ray diffraction pattern for the as-cast and heat treated Al_4_CrFeMnTi_0.25_ alloy.

**Figure 14 materials-16-05067-f014:**
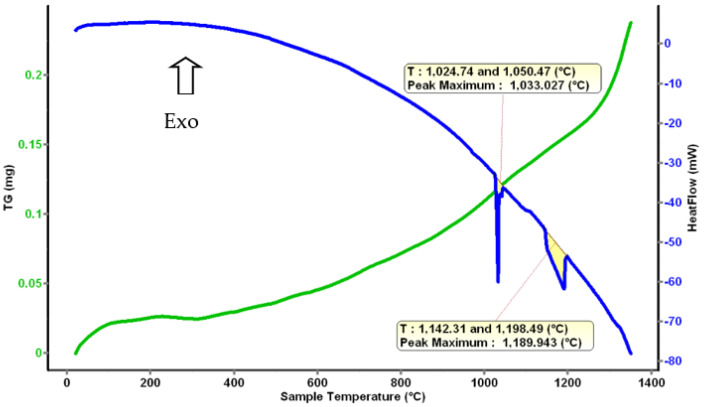
Thermal Analysis diagram for Al_4_CrFeMnTi_0.25_.

**Figure 15 materials-16-05067-f015:**
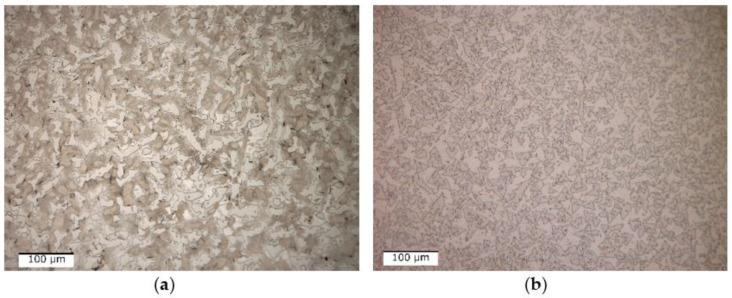
Optical micrographs of as-cast (**a**) and quenched (**b**) AlFeMnSi alloy.

**Figure 16 materials-16-05067-f016:**
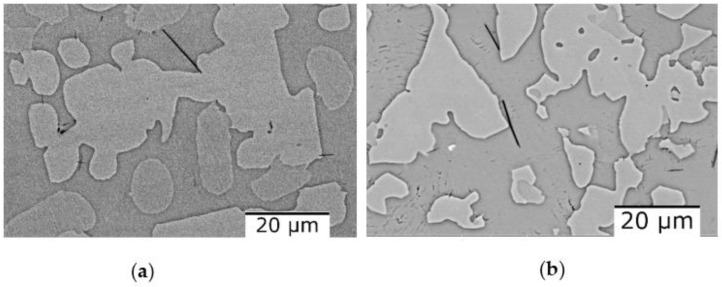
SEM-EDS image of the AlFeMnSi alloy in as-cast (**a**) and quenched (**b**) states.

**Figure 17 materials-16-05067-f017:**
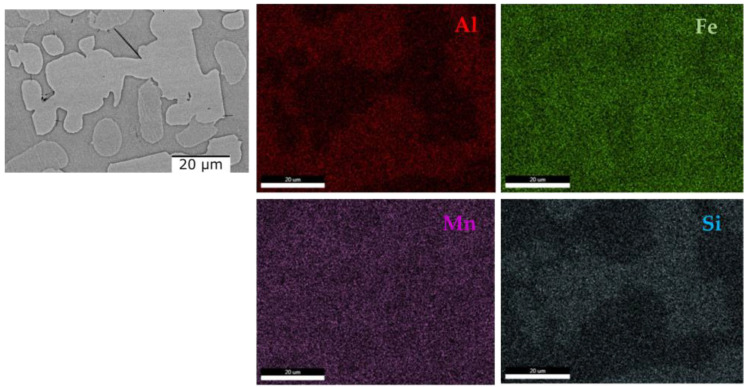
EDS mapping of the as-cast AlFeMNSi alloy.

**Figure 18 materials-16-05067-f018:**
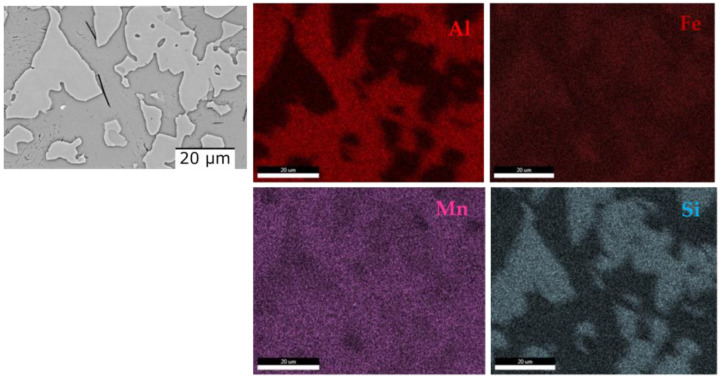
EDS mapping of the quenched AlFeMnSi alloy.

**Figure 19 materials-16-05067-f019:**
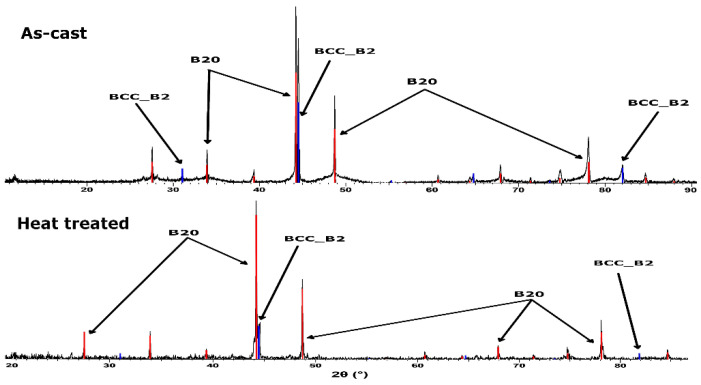
X-ray diffraction pattern for the as-cast and heat treatment AlFeMnSi alloy.

**Figure 20 materials-16-05067-f020:**
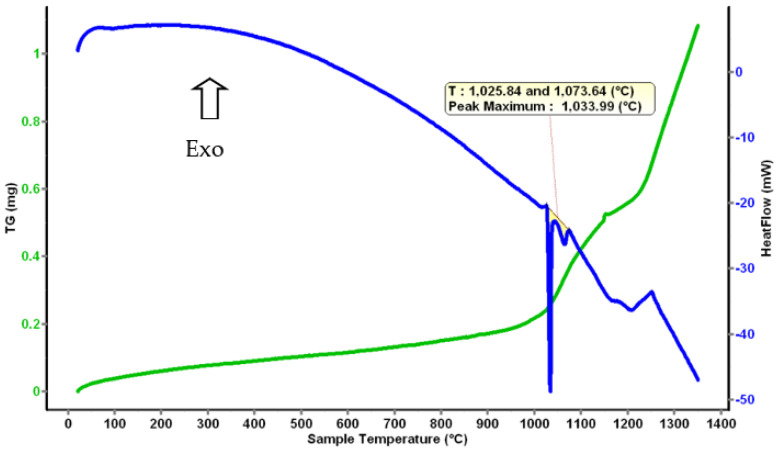
Thermal Analysis diagram for the AlFeMnSi alloy.

**Table 1 materials-16-05067-t001:** Criteria calculation results for AlCrFeMnTi system.

No.	Alloy	∆χ_Allen_ %	Λ (J/mol·K)	ρ (g/cm^3^)	Melting Temperature (°C)
1	Al_0.5_CrFeMnTi	9.35	0.29	6.01	1474.48
2	AlCrFeMnTi	8.90	0.30	5.62	1393.06
3	Al_1.5_CrFeMnTi	8.51	0.30	5.31	1326.45
4	Al_2_CrFeMnTi	8.17	0.31	5.06	1270.94
5	Al_2.5_CrFeMnTi	7.87	0.31	4.85	1223.97
6	AlCr_0.5_CrFeMnTi	9.39	0.31	5.47	1341.52
7	AlCrFeMnTi	8.90	0.30	5.62	1393.06
8	AlCr_1.5_FeMnTi	8.48	0.29	5.74	1435.24
9	AlCr_2_FeMnTi	8.12	0.28	5.84	1470.39
10	AlCr_2.5_FeMnTi	7.80	0.28	5.93	1500.13
11	AlCrFe_0.5_MnTi	8.81	0.31	5.41	1377.29
12	AlCrFeMnTi	8.90	0.30	5.62	1393.06
13	AlCrFe_1.5_MnTi	8.87	0.28	5.79	1405.97
14	ACrlFe_2_MnTi	8.78	0.27	5.94	1416.72
15	AlCrFe_2.5_MnTi	8.65	0.27	6.07	1425.82
16	AlCrFeMn_0.5_Ti	9.14	0.26	5.44	1409.52
17	AlCrFeMnTi	8.90	0.30	5.62	1393.06
18	AlCrFeMn_1.5_Ti	8.66	0.32	5.76	1379.60
19	AlCrFeMn_2_Ti	8.42	0.35	5.89	1368.39
20	AlCrFeMn_2.5_Ti	8.19	0.36	6.00	1358.90
21	AlCrFeMnTi_0.5_	7.45	0.31	5.77	1362.52
22	AlCrFeMnTi	8.90	0.30	5.62	1393.06
23	AlCrFeMnTi_1.5_	9.76	0.29	5.50	1418.06
24	AlCrFeMnTi_2_	10.29	0.28	5.40	1438.89
25	AlCrFeMnTi_2.5_	10.63	0.28	5.32	1456.51

**Table 2 materials-16-05067-t002:** Criteria calculation results for AlFeMnSi system.

No.	Alloy	∆χ_Allen_ %	Λ (J/mol·K)	ρ (g/cm^3^)	Melting Temperature (°C)
1	Al_0.5_FeMnSi	5.48	0.19	4.83	1292.62
2	AlFeMnSi	6.21	0.17	4.54	1213.58
3	Al_1.5_FeMnSi	6.56	0.16	4.32	1152.11
4	Al_2_FeMnSi	6.73	0.16	4.15	1102.93
5	Al_2.5_FeMnSi	6.80	0.16	4.00	1062.69
6	AlFe_0.5_MnSi	6.61	0.15	4.18	1167.66
7	AlFeMnSi	6.21	0.17	4.54	1213.58
8	AlFe_1.5_MnSi	5.87	0.18	4.84	1249.29
19	AlFe_2_MnSi	5.58	0.19	5.08	1277.86
10	AlFe_2.5_MnSi	5.32	0.20	5.29	1301.24
11	AlFeMn_0.5_Si	6.61	0.15	4.21	1209.09
12	AlFeMnSi	6.21	0.17	4.54	1213.58
13	AlFeMn_1.5_Si	5.87	0.19	4.81	1217.07
14	AlFeMn_2_Si	5.58	0.20	5.03	1219.86
15	AlFeMn_2.5_Si	5.33	0.21	5.22	1222.15
16	AlFeMnSi_0.5_	5.79	0.20	4.98	1184.95
17	AlFeMnSi	6.21	0.17	4.54	1213.58
18	AlFeMnSi_1.5_	6.35	0.16	4.23	1235.85
19	AlFeMnSi_2_	6.36	0.15	3.99	1526.81
20	AlFeMnSi_2.5_	6.32	0.14	3.81	1268.24

**Table 3 materials-16-05067-t003:** Optimization results for AlCrFeMnTi and AlFeMnSi systems.

Nr	Optimised Composition							∆χ_Allen_ %	Λ (J/mol·K)	ρ (g/cm^3^)	Melting Temperature (°C)
	Alloy	at %									
		Al	Cr	Fe	Mn	Ti	Si				
1	Al_4_CrFeMnTi_0.25_	55	14	14	14	3	-	5.37	0.29	4.39	1061.42
2	AlFeMnSi	16	-	34	33	-	17	6.21	0.17	4.54	1213.58

**Table 4 materials-16-05067-t004:** The chemical composition of the as-cast alloys.

Alloy	Composition Type	Al	Cr	Fe	Mn	Ti	Si
Al_4_CrFeMnTi_0.25_	Nominal	38.20	18.39	19.75	19.43	4.23	-
	Experimental	39.04	20.94	14.97	21.51	3.54	-
AlFeMnSi	Nominal	16.3	-	33.7	33	-	17
	Experimental	18.58	-	34.14	28.19	-	19.09

**Table 5 materials-16-05067-t005:** Microhardness results for the obtained alloy.

	Specimen	HV
Al_4_CrFeMnTi_0.25_	as-cast	1131.10
	Annealed	1008.20
AlFeMnSi	as-cast	1164.10
	Annealed	1368.70

**Table 6 materials-16-05067-t006:** The corrosion parameters of the tested samples in 3.5 wt.% NaCl at 25 °C.

Samples	E_OCP_ (v)	Rp (Ω·cm^2^)	E_corr_ (V)	i_corr_ (A/cm^2^)	CR (mm/Year)
Grey cast iron	−0.622	29.68	−0.646	4.82 × 10^2^	0.334
Al_4_CrFeMnTi_0.25_ as cast	−0.504	1566.17	−0.478	13.88	0.012
Al_4_CrFeMnTi_0.25_ annealed	−0.575	-	−0.522	39.28	0.013
AlFeMnSi as cast	−0.531	502.13	−0.499	2.27 × 10^2^	0.164
AlFeMnSi annealed	−0.539	2422.26	−0.515	2.60	0.003

Rp = polarization resistance; Ecorr = corrosion potential; icorr = corrosion current density; CR = corrosion rate.

## Data Availability

Not applicable.

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
