# Peer review of "Characterization of Complex Concentrated Alloys and Their Potential in Car Brake Manufacturing"

_materials, 2023, doi:10.3390/ma16145067_

Round 1

Reviewer 1 Report

1. It is desirable to increase the signatures of figures 1 and 2.

2. Maybe Figure 1,c? (line 222)

3. Maybe Figure 2, A? (line 247)

4. Why was it impossible to calculate the values in the program specifically for the alloy Al4CrFeMnTi0.25?

5. Table 3 has moved to another page, which is inconvenient to read.

6. Figures 6 and 7 should be aligned in size.

7. Check the numbering of the figures after Figure 9.

8. The size scale is not visible in Figure 6* and 13*. Is there 100 microns?

9. Figure 7* is not signed.

It may be necessary to remove the lower part on the SEM image, because it can be misleading.

10. It is necessary to make a distinguishable dimensional scale on SEM images and EDS.

11. In Figure 12* and 19* there are no signatures visible, it is very difficult to see what is indicated there. If this is not important information, then it should be deleted.

12. There are some mistakes in the design of the list of references.

Author Response

We would like to express our deepest gratitude  for evaluating our work and provide useful comments for its improvement. We also thank you for your expert opinion and your valuable comments. Below you will find our response to all the comments.

     1. It is desirable to increase the signatures of figures 1 and 2.

Answer: The text was increased in size.

  1. Maybe Figure 1,c? (line 222)

Answer: It was resolved

  1. Maybe Figure 2, A? (line 247)

Answer: It was resolved

  1. Why was it impossible to calculate the values in the program specifically for the alloy Al4CrFeMnTi0.25?

Answer: If one is referring to the criteria calculation program, a composition range was entered to determine element influence. Same for initial MatCalc simulation. But after the evaluation was presented a phase diagram was calculated for the alloy.

  1. Table 3 has moved to another page, which is inconvenient to read.

Answer: This is the review version of the manuscript the final version will be produced after editorial review. Nevertheless, the table was moved for better view.

  1. Figures 6 and 7 should be aligned in size.

Answer: It is resolved

  1. Check the numbering of the figures after Figure 9.

Answer: It is resolved. Also for tables.

  1. The size scale is not visible in Figure 6* and 13*. Is there 100 microns?

Answer: The scale was redone.

  1. Figure 7* is not signed.

Answer: It is resolved.

   It may be necessary to remove the lower part on the SEM image, because it can be misleading.

Answer: The lower part of the SEM contains also the scale, which is the main value in the microstructure.

  1. It is necessary to make a distinguishable dimensional scale on SEM images and EDS.

Answer: It was redone

  1. In Figure 12* and 19* there are no signatures visible, it is very difficult to see what is indicated there. If this is not important information, then it should be deleted.

Answer: It was redone. Thank you!

  1. There are some mistakes in the design of the list of references.

Answer: The references were reformatted and improved.

Reviewer 2 Report

Expert Review on the article titled " Characterisation of complex alloys with potential in car brake manufacturing"

1)      Overall, the abstract provides a concise overview of the article's content. However, there are a few suggestions and remarks I would like to make:

·         The sentence "There is a tremendous interest in the development of a material that has high strength, good heat transfer, corrosion resistance, and low density" could be improved by providing a brief explanation of why these properties are important for brake disc materials. This would help readers understand the significance of the research.

·         It would be beneficial to mention the specific application or purpose of the brake disc material early on in the abstract. This would give readers a clear context for the research and its potential impact.

·         The mention of "Complex concentrated alloys (CCA)" could be expanded upon. It would be helpful to provide a brief explanation or definition of CCA to ensure readers are familiar with the term.

·         The sentence "Several compositions were studied through thermodynamic criteria calculations and CALPHAD modeling, in order to determine appropriate structures" is clear, but it could be strengthened by briefly explaining what thermodynamic criteria calculations and CALPHAD modeling entail. This would enhance the readers' understanding of the methodology used in the study.

Overall, the abstract provides a good overview of the research conducted on the development of brake disc materials using complex concentrated alloys. By addressing the suggestions mentioned above, you can enhance the clarity and impact of the abstract.

2)      The introduction provides a comprehensive overview of the importance of brake disc materials and the challenges associated with their properties. Here are some observations and suggestions:

·         In the sentence "Braking system is a vital component of a vehicle and has the role of stopping or slowing down the moving vehicle by applying an artificial frictional resistance to the moving elements [1]," consider rephrasing it to make it more concise and clear. For example, "The braking system plays a crucial role in stopping or slowing down a vehicle by creating frictional resistance [1]."

·         In the sentence "Due to the importance and necessity of the brake disc, the material requirements have to be established and defined very clearly," it would be helpful to briefly explain why the brake disc is important and what role it plays in the braking system. This would give readers a better understanding of its significance.

·         The mention of lightweight and low particulate emissions as additional competitive factors in the automotive industry is relevant and highlights the need for improved brake disc materials. However, consider expanding on why these factors are important and how they impact the overall performance and environmental aspects of vehicles.

·         The description of various materials used in brake disc manufacturing is informative. However, it would be helpful to provide a brief explanation or examples of the advantages and disadvantages of each material to emphasize the need for alternative options.

·         The introduction mentions the environmental disadvantages of using grey cast iron for brake discs, such as high density, low corrosion resistance, fuel consumption, maintenance intervals, and particulate pollution. Consider providing specific data or statistics to support these claims and further emphasize the environmental impact.

·         The introduction briefly discusses complex concentrated alloys (CCAs) as a promising solution for brake disc materials. Consider providing a concise definition or explanation of CCAs to ensure readers understand the concept.

·         The introduction mentions previous research on alloy compositions and their properties, but it would be beneficial to provide a brief summary of the findings from these studies. This would help establish the current state of knowledge and highlight the research gap that the present study aims to address.

·         The objectives of the present study are clearly stated, including the focus on low-density alloys with specific properties, the comparison between two alloy systems, the investigation of heat treatment effects, and the potential replacement of gray cast iron. These objectives provide a clear roadmap for the research and its intended contributions.

Overall, the introduction provides a solid foundation for the research topic. By addressing the suggestions mentioned above, you can further enhance the clarity, relevance, and impact of the introduction.

3)      The "Materials and Methods" section provides a detailed description of the steps followed in the study, which provides a clear understanding of the experimental methods used. The authors considered various material selection criteria, such as density, cost, and melting temperature, to choose the constituent elements of the alloys. They also used parameters such as Allen's electronegativity and the geometric parameter Λ to analyze the formation of solid solutions in alloys. Thermodynamic and kinetic simulation of alloy structures using Matcalc Pro software is a sound approach to optimizing CCA alloy systems. CALPHAD modeling methods and the study of solid-state phase transformations are valuable tools for evaluating the properties of alloys. The characterization techniques used, such as inductively coupled plasma spectrometry (ICP-OES), scanning electron microscopy (SEM-EDAX), X-ray diffraction (XRD) and corrosion tests, are appropriate to assess the chemical composition, structure, mechanical properties and corrosion resistance of alloy samples. In summary, this section presents thorough and appropriate methods to select, analyze and characterize the alloys under study. The different experimental steps are described in a clear manner, which allows readers to understand and evaluate the results obtained.

4)      According to the results and discussions presented, there seems to be a correlation between the concentration of the elements and the formation of the phases in the alloys studied. The simulations carried out with the Matcalc program made it possible to observe the phase variations according to the variations in the concentration of the elements. In the AlCrFeMnTi alloy system, it was observed that the aluminum concentration influences the formation of the BCC_B2 phase, while the chromium concentration influences the formation of the BCC_A2 phase and the Cr3Mn5 intermetallic. Similarly, the iron concentration influences the proportion of the BCC_B2 phase. The presence of manganese promotes the formation of the BCC_B2 phase and the FeTi phase. Finally, the titanium concentration influences the variation of the BCC_B2 and H_Sigma phases. For the AlFeMnSi alloy system, the aluminum concentration influences the formation of the Al2Fe phase and the transition to the BCC_B2 phase. The presence of iron has a significant effect on the stability of the FeSi phase, while the manganese concentration influences the stability of the BCC_B2 and FeSi phases. Silicon has a strong impact on the formation of the FeSi phase and the BCC_B2 phase. With regard to the thermodynamic and kinetic criteria, the analysis of the different alloy compositions made it possible to determine the optimal composition for each alloy system. Criteria such as Allen electronegativity difference (Δχ), structure parameter Λ, density and melting temperature were taken into account. It has been observed that the concentration of aluminum reduces the density, while the concentration of titanium decreases the density and improves the resistance to high temperatures. Chromium and iron concentrations increase the melting temperature but have a negative effect on the formation of solid solutions. The manganese concentration has a positive influence on the formation of solid solutions, but increases the density and reduces the melting temperature.

·         it would be interesting to deepen the analysis of the mechanical properties, the resistance to corrosion and the performance of the alloys studied. In addition, practical experiments could be carried out to validate the results of the simulations. A more detailed analysis of the phase formation mechanisms and their influence on the properties of the alloys could also be carried out.

Overall, the manuscript presents a comprehensive and well-executed study that significantly contributes to the existing literature in the field. The authors have successfully addressed the research question and provided compelling evidence to support their findings. The methodology employed is rigorous, and the data analysis is sound. The results are clearly presented and effectively demonstrate the significance of the study. However, there are a few areas that require minor revisions to further strengthen the manuscript. These include clarifying certain sections for improved readability, addressing some minor inconsistencies in the data presentation, and expanding the discussion to provide more contextualization of the results. These revisions, once implemented, would enhance the overall quality of the manuscript. Therefore, I highly recommend acceptance with minor revisions.

Minor editing of English language required

Author Response

We would like to express our deepest gratitude  for evaluating our work and provide useful comments for its improvement. We also thank you for your expert opinion and your valuable comments. Below you will find our response to all the comments.

  • The sentence "There is a tremendous interest in the development of a material that has high strength, good heat transfer, corrosion resistance, and low density" could be improved by providing a brief explanation of why these properties are important for brake disc materials. This would help readers understand the significance of the research.

Answer: An explanation of properties meaning was added to the abstract.

  • It would be beneficial to mention the specific application or purpose of the brake disc material early on in the abstract. This would give readers a clear context for the research and its potential impact.

Answer: An initial description was added to the abstract.

  • The mention of "Complex concentrated alloys (CCA)" could be expanded upon. It would be helpful to provide a brief explanation or definition of CCA to ensure readers are familiar with the term.

Answer: A short definition of CCA was added to the text.

  • The sentence "Several compositions were studied through thermodynamic criteria calculations and CALPHAD modeling, in order to determine appropriate structures" is clear, but it could be strengthened by briefly explaining what thermodynamic criteria calculations and CALPHAD modeling entail. This would enhance the readers' understanding of the methodology used in the study.

Answer: Criteria parameters were added in parenthesis in the abstract.

2)      The introduction provides a comprehensive overview of the importance of brake disc materials and the challenges associated with their properties. Here are some observations and suggestions:

  • In the sentence "Braking system is a vital component of a vehicle and has the role of stopping or slowing down the moving vehicle by applying an artificial frictional resistance to the moving elements [1]," consider rephrasing it to make it more concise and clear. For example, "The braking system plays a crucial role in stopping or slowing down a vehicle by creating frictional resistance [1]."

Answer: The phrase was replaced

  • In the sentence "Due to the importance and necessity of the brake disc, the material requirements have to be established and defined very clearly," it would be helpful to briefly explain why the brake disc is important and what role it plays in the braking system. This would give readers a better understanding of its significance.

Answer: A paragraph explaining the functionality and conditions of disk brakes was added

  • The mention of lightweight and low particulate emissions as additional competitive factors in the automotive industry is relevant and highlights the need for improved brake disc materials. However, consider expanding on why these factors are important and how they impact the overall performance and environmental aspects of vehicles.

Answer: The particle pollution aspect was presented in the manuscript and with citations from recent and respectable articles.  A new paragraph was added to explain in depth the particulate pollution matter.

  • The description of various materials used in brake disc manufacturing is informative. However, it would be helpful to provide a brief explanation or examples of the advantages and disadvantages of each material to emphasize the need for alternative options.

Answer: The main material that is nowadays used in brake disk manufacturing is grey cast iron. the main disadvantages of the material is density and particulate emissions. Other materials are not widely used as they are much more expensive and/or have certain deficiencies (carbon fiber, composite materials, titanium alloys, etc.).

  • The introduction mentions the environmental disadvantages of using grey cast iron for brake discs, such as high density, low corrosion resistance, fuel consumption, maintenance intervals, and particulate pollution. Consider providing specific data or statistics to support these claims and further emphasize the environmental impact.

Answer: New text and references were added for supporting the affirmation.

  • The introduction briefly discusses complex concentrated alloys (CCAs) as a promising solution for brake disc materials. Consider providing a concise definition or explanation of CCAs to ensure readers understand the concept.

Answer: A definition of CCAs was presented already in the manuscript. Explanation was added for a better understanding. More details are offered in the cited reference.

  • The introduction mentions previous research on alloy compositions and their properties, but it would be beneficial to provide a brief summary of the findings from these studies. This would help establish the current state of knowledge and highlight the research gap that the present study aims to address.

Answer: A paragraph explaining the need for new materials was added in the introduction before the presentation of the present work.

Reviewer 3 Report

Materials-2488599

Title: Characterisation of complex alloys with potential in car brake  manufacturing

Reviewer Comments

1. In Abstract, include the application of chosen alloy, Method and results (Values).

2.  How can the uniform distribution of alloys?

3. The recent literature survey includes in the introduction part from (2022-2023)

4. In methodology, include the process chart diagram i. e. manufacturing steps as followed.

5. To make the table and tableted the parameters used in manufacturing process.

6. What type of microstructure is obtained from all the alloys, and why?

7. In conclusion, can we justify that the A concentration of Cr and Fe higher than 20 wt.% leads to a reduction in the proportion of BCC_B2 solid solution.

8. In conclusion, should not be required the deep decision of results, Just make the outcomes of the alloys in the pointwise.

To improves the quality of English used in the entire manuscript. 

Author Response

We would like to express our deepest gratitude  for evaluating our work and provide useful comments for its improvement. We also thank you for your expert opinion and your valuable comments. Below you will find our response to all the comments.

  1. In Abstract, include the application of chosen alloy, Method and results (Values).

Answer: The abstract contains already the type of alloys, the alloys application, structural characteristics and properties with values. Additional explanations were added to better describe the application

  1. How can the uniform distribution of alloys?

Answer: I am afraid I do not understand the question. The uniform distribution of phases can be observed in the microscopy results

  1. The recent literature survey includes in the introduction part from (2022-2023)

Answer: The references section contains 2 articles from 2022. One was added lately. No more recent references related specifically to the subject were found.

  1. In methodology, include the process chart diagram i. e. manufacturing steps as followed.

Answer: In general a process diagram chart is not presented for melting, casting and annealing processes unless require special operations. But if imperiously needed we can added to the second revision.

  1. To make the table and tableted the parameters used in manufacturing process.

Answer:  The process methods and parameters were presented in the Methods description section

  1. What type of microstructure is obtained from all the alloys, and why?

Answer: The microstructure of the alloys were presented through the characterization results already. Also discussions related to the possible justification for appearance were mentioned.

  1. In conclusion, can we justify that the A concentration of Cr and Fe higher than 20 wt.% leads to a reduction in the proportion of BCC_B2 solid solution.

Answer: The research work is offering an evaluation of the simulation results, which concludes that Cr and Fe percentage increase would hinder the formation of solid solutions in the AlCrFeMnTi system. 

  1. In conclusion, should not be required the deep decision of results, Just make the outcomes of the alloys in the pointwise.

Answer: As the manuscript has substantial analysing data we found that a larger than usual conclusion section is more appropriate.